# Perceptually relevant remapping of human somatotopy in 24 hours

James Kolasinski[1,2]*, Tamar R Makin[1,3], John P Logan[1], Saad Jbabdi[1], Stuart Clare[1], Charlotte J Stagg[1,4], Heidi Johansen-Berg[1]

[1]Oxford Centre for fMRI of the Brain, Nuffield Department of Clinical Neurosciences, University of Oxford, Oxford, United Kingdom; [2]University College, Oxford, United Kingdom; [3]Institute of Cognitive Neuroscience, University College London, London, United Kingdom; [4]Oxford Centre for Human Brain Activity, Department of Psychiatry, University of Oxford, Oxford, United Kingdom

**Abstract** Experience-dependent reorganisation of functional maps in the cerebral cortex is well described in the primary sensory cortices. However, there is relatively little evidence for such cortical reorganisation over the short-term. Using human somatosensory cortex as a model, we investigated the effects of a 24 hr gluing manipulation in which the right index and right middle fingers (digits 2 and 3) were adjoined with surgical glue. Somatotopic representations, assessed with two 7 tesla fMRI protocols, revealed rapid off-target reorganisation in the non-manipulated fingers following gluing, with the representation of the ring finger (digit 4) shifted towards the little finger (digit 5) and away from the middle finger (digit 3). These shifts were also evident in two behavioural tasks conducted in an independent cohort, showing reduced sensitivity for discriminating the temporal order of stimuli to the ring and little fingers, and increased substitution errors across this pair on a speeded reaction time task.

*For correspondence: james. kolasinski@ndcn.ox.ac.uk

## Introduction

Evidence for experience-dependent plasticity in the adult mammalian brain exists across a variety of sensory modalities (*Fu and Zuo, 2011*). The ordered somatotopy of primary somatosensory cortex (SI) has long served as a model system for studies of cortical reorganisation, with a wealth of evidence from both the non-human primate and rodent literature linking both extreme and subtle peripheral changes in somatosensory inputs over months or years to alterations in cortical maps (*Buonomano and Merzenich, 1998*; *Feldman and Brecht, 2005*).

In the human brain, there has also been evidence of experience-dependent remapping in SI. Considerable emphasis has been placed upon cross-sectional differences in the cortical organisation of SI observed in specialist populations, such as musicians, or patients with sensorimotor dysfunction, such as focal dystonia (*Elbert et al., 1995*; *Bara-Jimenez et al., 1998*; *Nelson et al., 2009*; *Kalisch et al., 2009*). However, only limited longitudinal evidence exists that directly demonstrates remapping at the level of human SI, either in response to altered hand usage patterns (*Stavrinou et al., 2007*) or more intensive Hebbian co-activation paradigms delivering specific patterns of tactile stimulation to the fingertips (*Pleger et al., 2001*, *2003*; *Hodzic et al., 2004*; *Vidyasagar et al., 2014*). There remains a limited understanding of the speed of SI plasticity and how cortical changes relate to behaviour. Here we address this gap in the literature, investigating the propensity for rapid experience-dependent cortical reorganisation and the behavioural relevance thereof.

Using a well-validated paradigm of single-subject fMRI mapping of human SI at 7 tesla (*Sanchez-Panchuelo et al., 2010*; *Kolasinski et al., 2016*), we asked two questions. First, can experience-

**eLife digest** The areas of the brain that receive inputs from our senses have a map-like structure. In an area called the visual cortex this map represents our field of vision; in the auditory cortex, it represents the range of different tones we can hear. The sense of touch is processed in the somatosensory cortex: an area of the brain that is organised around a map of the body, with adjacent regions of the cortex representing adjacent regions of the body. The clear structure of these brain regions makes them ideal for exploring how the organisation of the brain changes over time.

How quickly can changes to the touch inputs that the brain receives cause the map in the somatosensory cortex to reorganise? Can these effects be produced in just 24 hours? And would this remapping affect how we perceive touch? To investigate these questions, Kolasinski et al. glued together the right index and right middle fingers of healthy human volunteers. This separated the middle and ring fingers: a pair that usually move together due to the anatomical structure of the hand.

Functional magnetic resonance imaging of the brain's activity revealed that within 24 hours of the gluing, the brain's representation of the ring finger moved away from that of the middle finger, and towards the representation of the little finger. A perceptual judgment task mirrored this finding: after 24 hours of gluing, the participants became better at distinguishing between the middle and ring fingers and worse at distinguishing between the ring and little fingers. This is a powerful demonstration of the human brain's potential to adapt and reorganise rapidly to changes to sensory inputs.

The sense of touch declines gradually with age and may also be reduced as a result of disease such as stroke. A long-term challenge is to understand how the sensory regions of the brain change during this loss of sensation. Further research could then investigate how to maintain the structure of the cortical map to prolong or restore high quality touch sensation.

dependent plastic remapping of SI somatotopy be elicited in the human cortex in just 24 hr? Second, are any observed cortical changes reflected in altered tactile function? A complementary combination of fMRI and behavioural psychophysics were used to investigate the effect of a 24 hr manipulation in which the right index (D2) and right middle digit (D3) were joined together using skin glue. Based on the primate literature (*Clark et al., 1988*; *Allard et al., 1991*), we hypothesised that forced co-use of the glued digit pair would result in increased tactile co-activation across the digits, and an increase in their shared cortical representation. An alternative hypothesis posited that compensatory behaviour during the 24 hr gluing manipulation would promote changes in the cortical representations of adjacent, but unaffected digits.

In the fMRI cohort, SI digit somatotopy of the right hand was mapped at 7 tesla (*Kolasinski et al., 2016*) after two periods of normal hand use (control 1 and control 2), and after the gluing manipulation. In each session, fMRI data were acquired during a block design task and phase-encoding task. We first asked whether the amount of cortical overlap between digit representations in SI changed after the gluing manipulation in comparison with two control sessions using the phase-encoding fMRI data. Measures of inter-digit overlap were calculated for adjacent digit pairs (D2-D3: index-middle, D3-D4: middle-ring, D4-D5: ring-little; *Figure 1A*). A two-way repeated measures ANOVA indicated a significant interaction between session and digit pair on the amount of cortical overlap ($F_{(4,32)}$ = 13.412, p<0.0005, $\eta^2$:0.626) (*Figure 1B*). This was driven by a significant reduction in the cortical overlap between D3-D4 (Simple main effect: $F_{(2,16)}$ = 23.379, p<0.0005; Pairwise Sidak-corrected p<0.05) and a significant increase in the overlap of D4-D5 (Simple main effect: $F_{(2,16)}$ = 13.384, p<0.0005; Pairwise Sidak-corrected p<0.05) in the glued condition compared with both control sessions (*Figure 1B*). No significant change was found in the overlap between D2 and D3, the glued digits, where changes have been observed in similar but longer term studies in non-human primates (*Clark et al., 1988*; *Allard et al., 1991*). No shift in peak-to-peak distance between the digit representations (*Supplementary file 1*) or the overall surface area of each digit representation was observed (*Figure 1—figure supplement 1*). No systematic difference in the fit

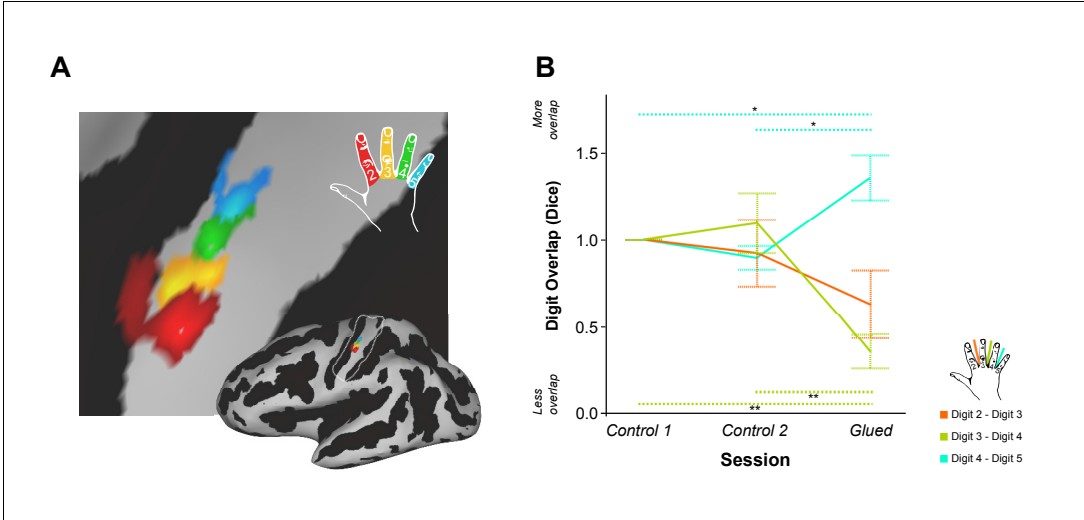

**Figure 1.** Patterns of rapid experience-dependent remapping in human SI. (**A**) An example digit map from an individual participant and session, shown on an inflated brain surface (bottom right), and a zoomed panel (Threshold FDR α = 0.01); sulci and gyri are shown in dark and light grey, respectively. The region of interest used for cortical overlap analysis is indicated by a white line on the inflated brain. (**B**) Cortical overlap between pairs of digit representations showed a significant reduction between D3 and D4, and a significant increase between D4 and D5 after the gluing manipulation compared with the two control conditions. *p<0.05 **p<0.005 Sidak corrected. Data in (**B**) are presented normalised to control one session; all statistics were performed on raw un-normalised data. Dice: Dice coefficient; Error bars represent standard error of mean.

The following source data and figure supplements are available for figure 1:

**Source data 1.** Data presented in *Figure 1B*.

**Figure supplement 1.** Cortical surface area of individual digit representations used in calculation of DICE coefficient at each time point.

**Figure supplement 2.** Overview of the fMRI experiment and the behavioural experiment study design.

**Figure supplement 3.** Data presented in *Figure 1B* split according to the order of sessions.

---

between the phase-encoding models and fMRI signal were observed across sessions (*Supplementary file 2*).

To further explore the observed change in inter-digit overlap with no change in overall cortical surface area of each digit representation, the fMRI representations of digit 4 used to calculate the cortical overlap metrics (Dice coefficients) were visualised for each participant and session (*Figure 2*). This data revealed that the observed changes presented in *Figure 1B* were driven by an expansion in the representation of digit 4 adjacent to digit 5 and a corresponding contraction at the boundary with digit 3.

Representational similarity analysis conducted on the block design fMRI data also implicated changes in the representation of digit 4 (*Figure 3*). These data demonstrated a shift in the representation of D4 away from D3 and towards D5.

Given the fMRI data were strongly suggestive of changes in cortical overlap outside of the glued digit pair (D2-D3), we sought to more fully investigate the observed pattern of short-term reorganisation by assessing the behavioural correlates of the gluing manipulation. The same experimental design and gluing manipulation were applied in a separate cohort. Instead of an fMRI scan, nine participants undertook behavioural psychophysics tasks during each session. The first task involved temporal order judgment (TOJ), where pairs of rapid vibrotactile stimuli were applied to adjacent digit pairs of the right hand at varying interstimulus intervals. Participants judged which digit was stimulated first. A psychometric function was fitted to the resulting accuracy data, from which a metric of

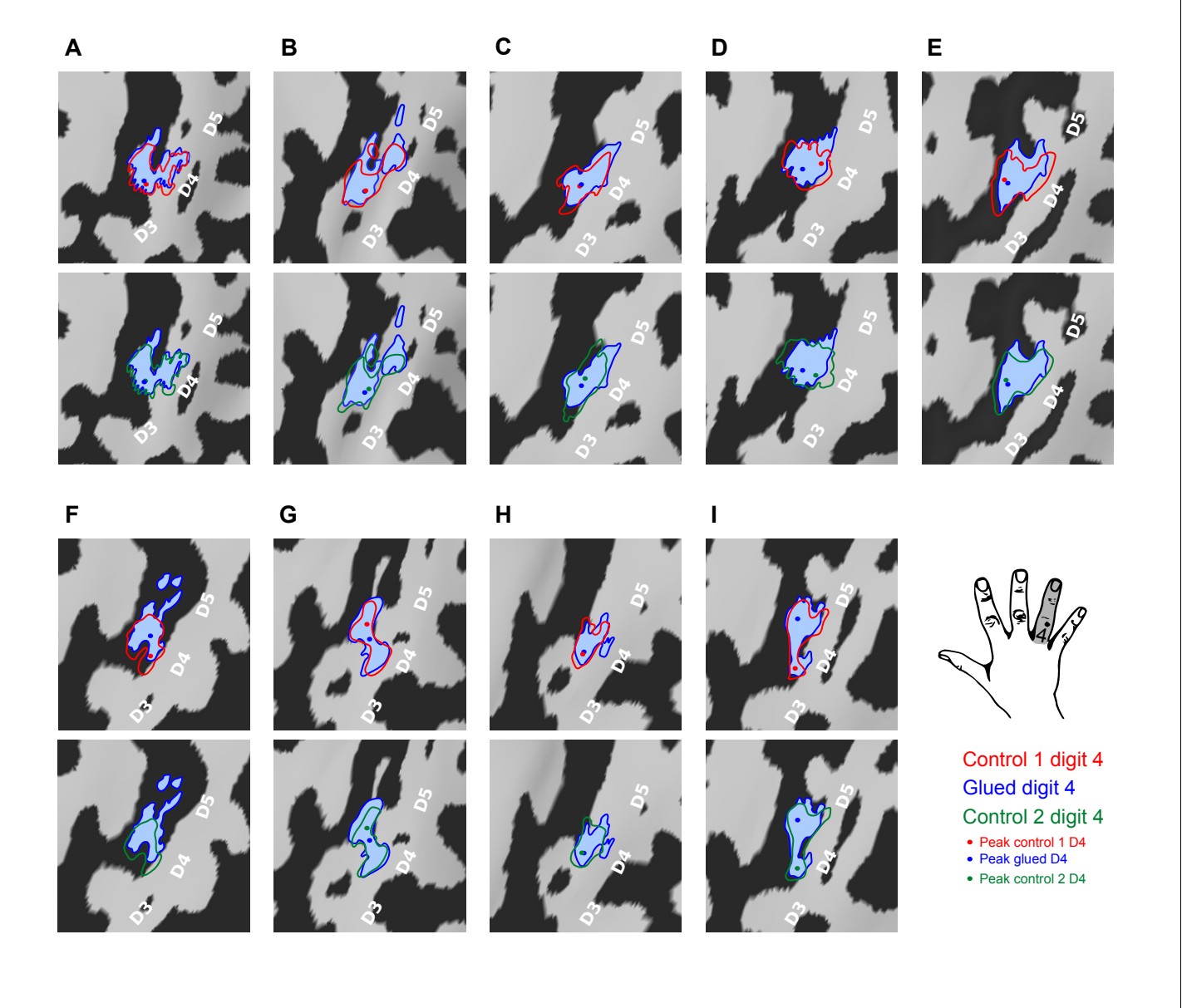

**Figure 2.** Pattern of shift in the cortical representation of digit 4 from inter-digit overlap analysis. Data displayed for all nine participants (A–I) showing the outline of the digit 4 representation mapped during the three sessions (Control 1, Control 2, Glued) overlaid on the individual participants' cortical surface reconstruction (sulci: dark grey, gyri: light grey; zoomed panel showing anatomical hand knob). The location of the peak vertex is shown with a coloured circle. These results demonstrated a consistent shift in the representation of digit 4 at the level of individual participants, such that while there is a minimal change in peak activation and area of the representation, the flank adjacent to the representation of digit 5 expands, and the flank adjacent to digit 3 contracts, consistent with the observed changes in cortical overlap (*Figure 1B*).

tactile discrimination ability was calculated for each adjacent digit pair (Just Noticeable Difference: JND). Greater values of JND are associated with poorer discrimination across an adjacent digit pair (*Figure 4A*). We asked whether the ability to distinguish the order of two stimuli delivered in rapid succession, one each on two adjacent digits, differed in the session directly following the gluing manipulation in comparison with the control sessions.

We generated behavioural predictions based on our fMRI findings of altered cortical maps after the gluing manipulation (*Figure 3*). Specifically, we hypothesised that the gluing manipulation would result in improved tactile discrimination (reduced JND) across D3-D4, which demonstrates reduced

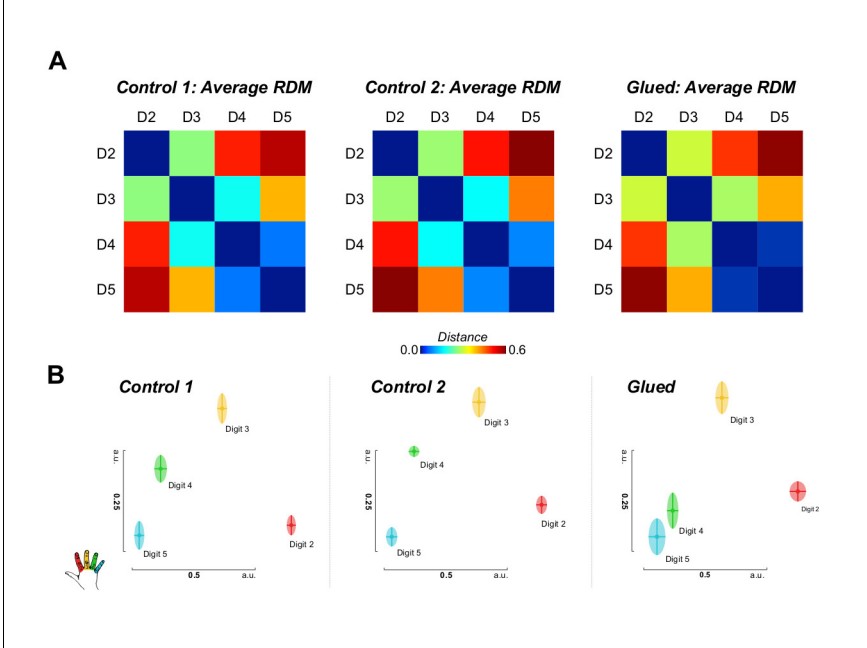

**Figure 3.** Representational similarity analysis of block design data yields complementary evidence of shift in the S1 representation of digit 4 away from digit 3 and towards digit 5. (**A**) Noise normalised parameter estimates from a standard GLM for each digit were used to construct representational dissimilarity matrices (RDMs) using Euclidean distance within a hand knob ROI derived individually for each subject from phase-encoding data from all sessions. Average raw distance values are shown for each session (**B**) Multidimensional scaling and Procrustes analysis of individual participants' distance matrices at each time point demonstrate schematically the observed shift in the representation of digit 4, away from digit 3, and towards digit 5, consistent with the observed pattern of cortical overlap and tactile discrimination changes. Two-way repeated measures ANOVA indicated a significant interaction between session and digit pair on the amount of cortical overlap ($F_{(2.0,16.2)}$ = 4.430, p=0.029, $\eta^2$:0. 356), driven by a shift in the representation of digit 4 away from digit 3 (Simple Main Effect: $F_{(2,16)}$ = 16.076; pairwise comparisons glued vs. control 1 and control 2: p<0.01). Multidimensional scaling yields the spatial relationship of representations in arbitrary units (a.u.).

The following source data is available for figure 3:

**Source data 1.** Data presented *Figure 3A*.

cortical overlap, and diminished tactile discrimination (increased JND) across D4-D5, which demonstrates increased cortical overlap. Data from eight participants met the goodness of fit threshold for all sessions ($R^2 > 0.4$). A two-way repeated measures ANOVA indicated a significant interaction between session and digit pair on tactile discrimination (JND) ($F_{(4,28)}$ = 14.613, p<0.0005, $\eta^2$:0.676). This was driven by a significant reduction in JND across D3-D4 (Simple main effect: $F_{(2,14)}$ = 14.631, p<0.0005; Pairwise Sidak-corrected p<0.05), and a significant increase in the JND across D4-D5 (Simple main effect: $F_{(2,14)}$ = 10.578, p=0.002; Pairwise Sidak-corrected p<0.05) in the gluing condition compared with both control sessions (*Figure 4B*). No significant change in tactile discrimination was found for D2-D3, the glued digits. In other words, after the gluing manipulation, tactile discrimination was improved across D3-D4 and was diminished across D4-D5, with no change in the glued digits D2-D3. This finding was supported by similar results from a second task, assessing motor confusion, which involved rapidly cued button presses using individual digits. There was an increase in the level of confusion between digits 4 and 5 after the gluing manipulation compared with the control conditions (*Figure 4—figure supplement 1*).

The fMRI and psychophysics results present complementary evidence for functionally relevant reorganisation in human SI following just a 24 hr peripheral change in hand use. Previous longitudinal studies of detailed somatotopic remapping in humans using MEG or standard resolution fMRI have provided evidence for a general change in the distance between digit representations

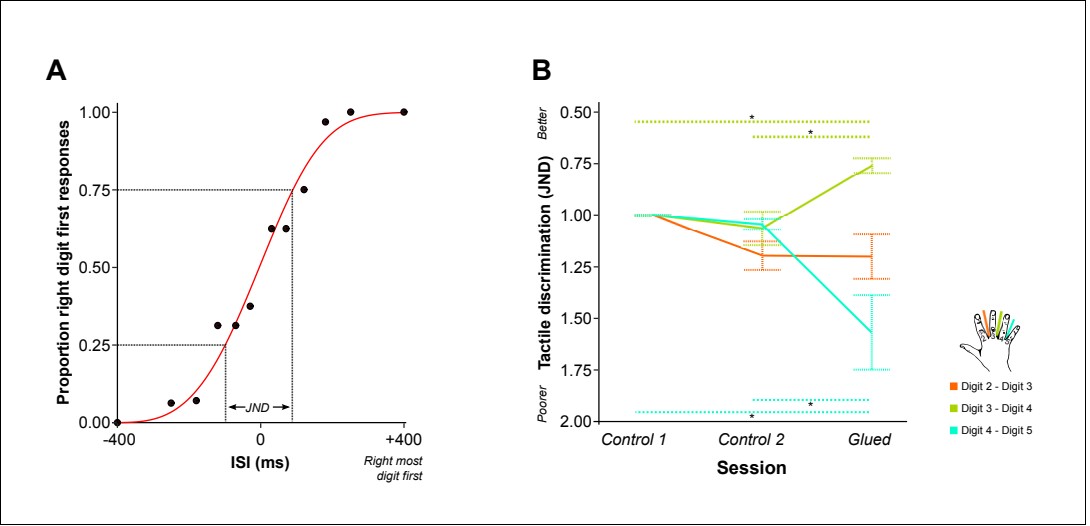

**Figure 4.** Patterns of rapid experience-dependent remapping in SI mirror peripheral changes in tactile discrimination performance. (**A**) An example of accuracy data from the temporal order judgment task. Accuracy scores (black dots) are illustrated for an individual participant and a single run assessing performance across D2-D3. Data are fitted with a logistic function (red line) from which the just noticeable difference (JND) is calculated: a measure of temporal tactile acuity; greater JND means poorer tactile discrimination. (**B**) Tactile discrimination improved significantly between D3 and D4, and worsened significantly between D4 and D5 after the gluing manipulation compared with the two control conditions. In summary, fMRI evidence of rapid cortical remapping (*Figure 3*) concurs with behavioural changes in tactile function (**B**), such that the digit pair with reduction in cortical overlap (D3–D4) also shows increases in tactile discrimination, whereas the digit pair showing increase in cortical overlap (D4–D5) demonstrates worsening of tactile discrimination *p<0.05 **p<0.005 Sidak corrected. Data in (**B**) are presented normalised to time point control 1; all statistics were performed on raw un-normalised data. ISI: inter-stimulus interval; JND: Just Noticeable Difference. Error bars represent standard error of mean.

The following source data and figure supplement are available for figure 4:

**Source data 1.** Data presented *Figure 4B*.
**Source data 2.** Data presented *Figure 4—figure supplement 1B*.
**Figure supplement 1.** In a motor confusion task involving rapid button presses using the four digits under study (D2, D3, D4, D5) there is an increase in the number of mis-presses between digits 4 and 5, consistent with the observed pattern of increased cortical overlap and representational similarity observed between digit 4 and 5.

---

(*Stavrinou et al., 2007*; *Mogilner et al., 1993*), or a synchronisation in activity pattern of non-adjacent digits (*Vidyasagar et al., 2014*) after coupled stimulation across digits. Conversely, an intervention involving hand and arm immobilisation drove diminished tactile acuity and a corresponding reduction in BOLD activity in contralateral SI (*Lissek et al., 2009*). Our results build on this work, demonstrating that usage-dependent changes in the overlap of SI digit representations are associated with a corresponding change in the ability to differentiate tactile inputs to the digits. Specifically, an increase in the cortical overlap between D4 and D5 was accompanied by a reduction in the tactile discrimination ability across these two digits.

A consistent feature of our findings was that cortical reorganisation and behavioural change was not observed for the glued digit pair (D2-D3). Instead, we saw changes in cortical overlap, and corresponding change in behavioural performance, for the other digit pairs. Specifically, we saw a shift of D4 away from D3 and towards D5. These findings reject our hypothesis of increased tactile synchronisation driving an increase in cortical overlap. Instead, these data support the alternative hypothesis that compensatory behaviour in other digits during the 24 hr manipulation drives off-target effects.

In light of the considerable dexterous abilities specific to the human hand (*Young, 2003*) and the short duration of the gluing manipulation, it is feasible that the observed remapping reflects

compensatory behavioural changes in the pattern of hand use during the manipulation. Rather than learning to coordinate and co-use the glued digits, participants may have adapted synergies with the non-manipulated digits. In this case, the increased physical separation of D3 and D4 during the gluing reduces the usually high degree of anatomical enslavement seen peripherally between these two digits (*Kim et al., 2008*), freeing D4 to function even more synergistically with D5. This argument is supported by the observed pattern of increased motor confusion between digits 4 and 5 after the gluing manipulation (*Figure 4—figure supplement 1*), demonstrating potential changes in movement synergies, that may explain the somatosensory changes observed. This finding complements recent observations that patterns of synergistic digit usage are strongly reflected in the generalisation of tactile learning from trained to adjacent untrained digits (*Dempsey-Jones et al., 2016*).

The two fMRI datasets showed independent evidence of a shift in the digit 4 representation during the gluing manipulation: away from digit 3, and towards digit 5 (*Figures 1* and *3*), raising questions regarding the underlying mechanism driving this reorganisation. Similar studies undertaken over a period of months in non-human primates have reported the emergence of overlapping receptive fields in response to the fusion of two adjacent digits (*Clark et al., 1988*; *Allard et al., 1991*). The spatial resolution of BOLD fMRI is, of course, poorer than invasive mapping techniques. Nonetheless, it is possible that the observed changes in cortical activation could result from shifts in population receptive fields at the boundary of cortical digit representations (*Besle et al., 2014*), or even changes in multidigit receptive fields (*Thakur et al., 2012*). As well as reorganisation in the cortex, strong evidence also suggests considerable plastic potential in subcortical structures, which is mirrored in the cortex (*Jones, 1996*; *Rausell et al., 1998*; *Kambi et al., 2014*). The remapping observed in SI here may therefore result from reorganisation in lower level nuclei of the ascending somatosensory pathway.

A shift in the cortical representation of digit 4 (*Figure 2*), rather than an overall enlargement of the representation, distinguishes the observed pattern of cortical change from previous studies of artificial tactile co-activation, wherein a generalised cortical magnification of the manipulated digit is observed (*Pleger et al., 2001*, *2003*; *Hodzic et al., 2004*). The cortical shift perhaps reflects how naturalistic behavioural changes in hand use are reflected in the cortex, with remapping based on functional need; in this case, leading to perceptual improvements in certain digit pairs (D3/D4), and worsening in others (D4/D5).

What mechanism drives the observed change in cortical overlap? In vivo studies in rodent barrel cortex demonstrate that a 24 hr period of altered sensory input can induce increases in synaptic density in the corresponding cortical representation (*Knott et al., 2002*). This could unmask or potentiate pre-existing divergent or silenced connections between adjacent digit representations (*DeFelipe et al., 1986*; *Recanzone et al., 1992*; *Huntley, 1997*). The question remains: how could this cortical shift occur with no associated change in the surface area or location of peak activation of the digit 4 representation? Patterns of local horizontal connectivity in the cortex are established during development and moulded throughout life (*Sur and Rubenstein, 2005*). Homeostatic mechanisms also exist to maintain balance in cortical activity (*Turrigiano and Nelson, 2000*), and to preserve the scaling and pattern of cortical topography (*Sharma et al., 2000*; *Vanderhaeghen et al., 2000*). It is plausible that the gluing manipulation results in the potentiation of excitatory horizontal connectivity between the representations of digits 4 and 5, and the concurrent weakening of connectivity between the representations of digits 3 and 4. After 24 hr, we observe the resulting pattern of cortical reorganisation (*Figures 2* and *3*), wherein the relative positions of the representations have changed, but features of the maps firmly established in development, remain in place. A comparable process of reduced inhibition prior to remapping has been reported in studies of sensory deprivation in rodent barrel cortex. Reduced inputs after whisker clipping produce a local pattern of cortical disinhibition and broadening excitation, followed by subsequent cortical contraction of the representation (*Albieri et al., 2015*). In this case, changes in the relative inputs across different digits may prompt a transient disinhibitory cortical milieu, and the subsequent re-establishment of modified excitatory networks within the cortex, wherein the cortical representations have shifted to reflect new usage patterns. The unmasking of latent excitatory connections prior to any homeostatic increase in lateral inhibition to rebalance cortical excitability could explain the observation of more marked changes in tactile perception and representational surface area after very short and intense tactile training interventions, or rTMS protocols (*Pleger et al., 2003*; *Tegenthoff et al., 2005*; *Ragert et al., 2008*; *Dinse and Tegenthoff, 2015*).

The consistent cortical changes at the flanks of the digit 4 representation (*Figure 2*) in the absence of changes in the location of peak activation suggest that the latter may not be the most informative feature of rapid map reorganisation. The existence of such hard-wired elements in cortical somatotopy is supported by evidence of SI constraining the extent of reorganisation within the somatosensory system: acting as a blueprint for lower level nuclei (*Zembrzycki et al., 2013*), enforcing some limits on plasticity, and potentially explaining the persistence of SI somatotopic features long after marked peripheral changes, such as amputation (*Kikkert et al., 2016*).

Given the increasingly recognised contribution of motor efference information to activity patterns in somatosensory cortex (*Lee et al., 2008*, *2013*), it is also possible that changes in motor outputs over the course of 24 hr may shape patterns of SI activity, promoting a top-down form of SI experience-dependent plasticity.

The rapidity of cortical changes shown herein supports further investigation of rapid and targeted therapeutic interventions in conditions such as focal dystonia, where maladaptive changes in somatotopy have previously been reported (*Bara-Jimenez et al., 1998*). The mechanistic link between short-term plasticity and long-term circuit changes remains unclear (*Holtmaat and Svoboda, 2009*; *Johansen-Berg et al., 2012*). Therefore, further questions remain as to how short-term peripheral manipulations could be used to induce sustained modification or refinement of fine grain functional cortical organisation, as a means to enhance or rehabilitate tactile function.

## Materials and methods

All data were acquired in accordance with local central university research ethics committee approval (University of Oxford MSD-IDREC-C2-2013-05). Eighteen participants were recruited (age range 19–33 years, eight females; *Supplementary file 3*) and provided written informed consent. All participants were right handed according to the Edinburgh Handedness Inventory (*Oldfield, 1971*), had no history of neurological or psychiatric illness, and met local MRI safety criteria where appropriate.

### fMRI experiment

#### Experimental design

Nine healthy adult volunteers participated in this experiment, in line with the closest previous studies of longitudinal remapping in human SI (*Pleger et al., 2003*; *Young, 2003*). Participants were scanned in four sessions. The order of experimental conditions is outlined in *Figure 1—figure supplement 2*. The first session chronologically (control 0) was treated as a habituation scan, to allow participants to get used to the unique aspects of being scanned in an ultra-high-field magnet, and the fMRI tasks. Included in the analysis were control 1 and control 2: two sessions that followed a period of normal hand use, and a glued session, which followed a 24 hr period of the gluing manipulation described above. The order in which the participants undertook the control and glued session was counterbalanced across the group.

During the gluing manipulation, the medial aspect of the right index digit and lateral aspect of the right middle digit were attached using a surgical grade cyanoacrylate skin glue (Derma+flex, Chemence Medical Products Inc., Alpharetta, GA), leaving the tips and pads of the digits exposed. The mixture octyl- and butyl-cyanoacrylates in this glue were selected for their combination of strength and flexibility. Participants were able to flex and extend the two attached digits and were instructed to go about the day-to-day routine as normal.

#### MRI data acquisition

Functional MRI data were acquired using a Siemens 7T Magnetom system with a 32-channel head coil. Blood oxygenation level dependent (BOLD) fMRI was acquired using a T2*-weighted multislice gradient echo planar imaging (EPI), with true axial slices centered on the left anatomical hand knob in the z-axis (TR 1500 ms, TE 25 ms, slice thickness 1.2 mm, in-plane resolution 1.2 × 1.2 mm, 22 axial slices, GRAPPA factor = 2). One single volume contrast matched EPI acquisition of the same resolution was also acquired for registration purposes. fMRI data acquisition, task and data preprocessing have been detailed previously (*Kolasinski et al., 2016*).

Structural MRI data were acquired using a 3T Siemens Trio system during one of the scan sessions with a multi-echo magnetisation prepared rapid acquisition gradient echo (MEMPRAGE) sequence

(*van der Kouwe et al., 2008*) (TR 2530 ms, TE 1.69, 3.55, 5.41 and 7.27 ms, slice thickness 1.0 mm, in-plane resolution 1.0 × 1.0 mm, GRAPPA factor = 2).

## fMRI tasks and stimuli

fMRI data were acquired during an active motor task involving visually cued movements of individual digits, from digit 2 (D2: index) to digit 5 (D5: little) inclusive. The digits involved in the gluing manipulation were released prior to any scan. During the tasks, participants made individual movements of D2, D3, D4 and D5 in the form of button presses on an MRI-compatible button-box (in-house manufactured) placed on the right thigh of the participant. Instructions were delivered via a visual display projected into the scanner bore. Four white circles were presented, representing the four fingers of the right hand. The circles flashed individually at a frequency of 1 Hz, instructing movements of a specific digit at that rate. During each visit (*Figure 1—figure supplement 2*), a block design task and a phase-encoding task were undertaken during fMRI acquisition.

### Phase-encoding fMRI task

The task followed a phase-encoding design, involving continuous button presses with no rest periods. The task involved 8 s blocks of movement of each digit. A phase-encoding forward task cycled through blocks of D2, D3, D4 and D5, with a total of eight repetitions of the cycle. A phase-encoding backward task cycled through blocks of D5, D4, D3 and D2, with a total of eight repetitions of the cycle. The total duration of task fMRI acquisition was 8 min 50 s.

### Block design fMRI task

A block design task was undertaken involving the same cued movements of a given digit in blocks of 12 s, contrasted with 12 s blocks of rest. Blocks of movement for each digit were repeated four times, giving a total of 16 movement blocks and 16 rest blocks. The total duration of the task was 6 min 30 s. Block order was counterbalanced, and different orders used across the different visits, counterbalanced across participants.

## MRI data preprocessing

fMRI data were processed using tools from the FMRIB Software Library (*Jenkinson et al., 2012*). All fMRI data were analysed at the level of individual participants and sessions; no group averaging was undertaken across session or across participants. fMRI data were subject to motion correction with FSL MCFLIRT (*Jenkinson et al., 2002*), the removal of non-brain tissue with the FSL BET (*Smith, 2002*), high-pass temporal filtering to remove slow drift in BOLD signal (100 s cut-off) and spatial smoothing (Gaussian kernel full-width half maximum 1.5 mm). fMRI data were subjected to exclusion in cases of visible spin history motion artifact as a result of sharp motion during one or more scan sessions (1 mm of absolute mean displacement in fewer than five volumes), as in previous studies (*Kolasinski et al., 2016* ); however, no fMRI data were excluded from this study.

## Phase-encoding fMRI analysis: inter-digit overlap

The phase-encoding fMRI data were used to map representations of individual digits in S1; using these representations, it is possible to derive a measure of overlap between the representations of adjacent digits. The phase-encoding task fMRI analysis has been outlined fully elsewhere (*Kolasinski et al., 2016*). The travelling wave approach described was applied to data derived from each individual participant from each scan session. In brief, this approach applies a cross-correlation with a number of iteratively time-shifted models to find a time point in the phase-encoding forward (D2-D5) and phase-encoding backward (D5-D2) cycle at which each voxel responded maximally. In this case, the reference model was a gamma-convolved boxcar: 8 s 'on' and 24 s 'off', repeated eight times. With each iterative shift of the model, a cross-correlation was calculated with the BOLD signal at each voxel, with sufficient shifts to cover the entire digit cycle from D2-D5 or D5-D2.

Each iteration of the model was assigned to a specific digit based on the position of the 'on' period in the digit cycle. The $r$-value maps for the lags of each digit were averaged to yield three-dimensional digit maps (D2, D3, D4 and D5) for the phase-encoding forward and backward tasks in each participant, in each session. These maps were resampled into the participant's high-saturation EPI space from the corresponding session using FSL FLIRT. The forward and backward $r$-value maps

were then averaged to yield a digit representation for each participant in each session. A corresponding z-statistic map was calculated for each of the four digit representations in each participant, in each session.

Digit maps from all sessions were resampled into the contrast matched EPI space of the control 0 habituation scan using FSL FLIRT (*Jenkinson and Smith, 2001*) (6 degrees of freedom, normalised correlation cost function) so as not to bias registration towards any individual session in the analysis. The control 0 contrast matched EPI image was registered to the structural MEMPRAGE image using boundary-based registration (*Greve and Fischl, 2009*) (BBR; Degrees of freedom: 6, FMRIB's Automated Segmentation Tool (FAST) white matter segmentation, no search), refined using the white matter and pial surfaces using blink comparison in Freeview.

Cortical surface reconstructions were derived from MEMPRAGE images using FreeSurfer (*Dale et al., 1999*). The z-statistic digit maps from each session were projected from each participant's raw BOLD fMRI EPI space onto the participant's cortical surface using Connectome Workbench (*Marcus et al., 2011*). Once projected to the cortical surface, each digit representation for each session was thresholded using FDR ($\alpha$ = 0.01), and binarised for subsequent overlap analysis.

A measure of cortical overlap between adjacent digits (D2/D3, D3/D4 and D4/D5) was calculated from the binarised digit representations for each participant and session using the Dice coefficient (*Dice, 1945*). The metric varies from a value of 0, indicating no overlap between the representations, to a value of 1, indicating perfect overlap between the representations. The Dice coefficient was calculated within a FreeSurfer-derived SI region of interest, overlapping with the BOLD fMRI EPI acquisition volume. Where A and B are the area of the two digit representations, the Dice Coefficient is expressed as:

$$\frac{2|A \cap B|}{|A| + |B|}$$

Patterns of peak activation for each digit during each session were calculated using Connectome Workbench, defined as the vertex with the peak z-statistic for each digit within the FreeSurfer-derived SI region of interest. The geodesic distance between adjacent digit peaks was subsequently calculated (*Supplementary file 1*).

## Block design fMRI: representational similarity analysis

Multivariate analysis was conducted using block design data to compare the representational structure of digit maps in the glued versus control sessions. fMRI data were analysed at the single-subject level using a standard GLM approach using FSL FEAT (*Jenkinson et al., 2012*), with an individual regressor for movement of each digit (D2, D3, D4, D5), using a gamma-HRF boxcar (four 12 s blocks of activity per digit during the task). The resulting parameter estimates and residuals were used for subsequent calculation of representational dissimilarity matrices (RDMs) using the MRC CBSU toolbox (http://www.mrc-cbu.cam.ac.uk/methods-and-resources/toolboxes/) (*Nili et al., 2014*). Parameter estimates were subjected to univariate noise normalisation by voxelwise covariance estimates. A region of interest was defined for each participant using the phase-encoding data, averaging across sessions, to yield a region of activation (FDR $\alpha$ = 0.01) in the anatomical hand knob. The noise-normalised parameter estimates underlying the ROI were then used to construct representational dissimilarity matrices using Euclidean distance for each session. The distance metrics for adjacent digit pairings were subjected to a two-way repeated measures ANOVA, with one factor: session (Control 1, Control 2, Glued) and a second factor: digit pair (D2/D3, D3/D4, D4/D5). For visualisation purposes, the individual RDMs for each session were subjected to classical multidimensional scaling (MDS). Within each session, the results of the MDS for each participant were aligned using Procrustes analysis. The standard errors were corrected for the reduction in variability induced by the alignment, as described previously (*Ejaz et al., 2015*).

## Behavioural experiment

### Experimental design

Nine healthy adult volunteers participated in this experiment, none of whom participated in experiment one. Participants attended four behavioural testing sessions in total. These sessions were scheduled in the same way as the scans the fMRI Experiment (*Figure 1—figure supplement 2*), with

an initial habituation session (Control 0). Included in the analysis were control 1 and control 2: two sessions that followed a period of normal hand use, and a glue session, which followed a 24 hr period of the gluing manipulation (*Figure 1—figure supplement 2*). The gluing manipulation was identical to that applied in the fMRI experiment, and was not in place during any behavioural testing. The order in which the participants undertook the control and glued condition was counterbalanced across the group.

## Temporal order judgment task

Participants performed a temporal order judgment (TOJ) task on vibrotactile stimuli presented on the distal pad of digits 2, 3, 4 and 5 of the right hand. This is a standard task used previously to study hand representation (*Shore et al., 2005*; *Azañón et al., 2015*). Participants sat in a testing room in a static chair placed a fixed distance from a table, with their arms resting on the table uncrossed. Their right hand was positioned in the midline, resting on a two-button response device (manufactured in-house). On each of the two buttons of the response device was mounted a vibrotactile stimulator delivering a fixed amplitude suprathreshold pulse (Two Oticon-A, 100 Ω bone conduction vibrators; 20 ms, 200 Hz sinusoidal; amplitude at least 10 times threshold), such that the ventral pads of adjacent digit pairs could be directly opposed to the flat surface of the stimulator. The relative position of the stimulators could be adjusted to accommodate different hand sizes and the relative lengths of different digit pairs. The view of the right hand and forearm was fully occluded and pink noise was used to mask any subtle auditory cues from the vibrotactile stimuli. Time of day was matched across conditions.

Participants performed a two alternative forced choice task following instructions on a laptop screen positioned at eye level at a distance of 56 cm. Participants performed three runs of this task, one for each digit pairing (D2-D3, D3-D4 and D4-D5; order counterbalanced across conditions and participants). Each run of the task was composed of 192 trials. During each trial, participants received one stimulus to each of the two digits and were asked to judge whether the left or right stimulus came first by pressing with the corresponding digit. Participants were cued on the screen with 'Ready' to alert of an oncoming stimulus, 'Response' to cue a choice and 'No response' in cases of failure to press respond within 2 s of the stimuli. Participants were instructed to fixate on a central fixation cross present on the screen at all times; monitoring was in place to ensure participants' eyes remained open at all times. The inter stimulus intervals (ISIs) were +400,–400,+250,–250,+180,–180, +120,–120,+70,–70,+30 or 30 ms where, by convention, a positive ISI denotes that the right digit was stimulated first. For each digit pair, the 12 ISIs were presented 16 times in a randomly assigned order.

## Temporal order judgment data analysis

For each run in each session, the proportion of right-first responses for each ISI was fitted to a logistic function (*Figure 4A*) using MATLAB and Statistics Toolbox Release 2014b (The MathWorks, Inc., Natick, MA). Data from one participant were excluded on the basis of a poor fit for digit pairs at one session ($R^2 < 0.4$), threshold selected from earlier study (*Azañón et al., 2015*). The fit of the remaining data far exceeded this threshold (Range: 0.68–0.99). In TOJ tasks, the just noticeable difference (JND) is defined as half of the difference between two ISIs that yield the 'right-most digit first' judgment in 75% and 25% of the trials. Smaller values of JND are therefore associated with better performance in terms of temporal tactile resolution.

## Motor confusion task

A second behavioural test involved a motor confusion task. This task involved rapid, visually cued movements of individual digits, in the form of presses on a four-button button box (in-house design). The task was delivered in the same testing environment as the TOJ task. Movements were cued every 700 ms, in blocks of 210 button presses, separated by a 20-s rest period, with a total of four blocks acquired in each session. The order of button presses was pseudorandomised, and the same button press was not repeated twice consecutively; the overall sequence in each block included an equal number of all possible adjacent digit pairings, including both neighbouring digits (e.g. D2 D3) and non-neighbouring digits (e.g. D3 D5). Participants were instructed to respond as quickly and accurately as possible to each cue and to not correct any perceived errors through double presses.

Data were analysed including only responses within the 700-ms response window. Responses were categorised as correct or incorrect; for the latter, the digit used to make the response was recorded.

## Statistics

Statistical analyses and graphing used JMP (Version 12.0, SAS Institute, Cary, NC) and Statistics Package for the Social Sciences (SPSS, Version 22.0, IBM Corporation, Armork, NY). The within-subjects data applied for the fMRI and psychophysics data were interrogated with independent two-way repeated measures ANOVAs, selected on the basis of data normality (Shapiro-Wilk confirmed). Significant interactions were further interrogated with analysis of simple main effects; full statistical outputs are provided in *Supplementary file 4*. All repeated measures ANOVAs were also run using a single value for the control condition by averaging the values from control A and control B; the same pattern of interactions and simple main effects were observed. Experimenters were not blinded to the condition (glued or control) during data acquisition for practical reasons.

## Acknowledgements

JK holds a Stevenson Junior Research Fellowship at University College, Oxford. TRM and CJS both hold Wellcome Trust/Royal Society Henry Dale Fellowships (TRM 104128/Z/14/Z; CJS: 102584/Z/13/Z). SJ holds an MRC Career Development Fellowship (MR/L009013/1). HJB holds a Wellcome Trust Principal Research Fellowship (110027/Z/15/Z). The work was additionally supported by the NIHR Oxford Biomedical Research Centre. Support for the 7T scanner was provided by the Medical Research Council. The authors thank Ellen Thomas, Maria Blöchl and Charles Spence for their assistance with piloting the behavioural paradigm, Harriet Dempsey-Jones and Lewis Gaul for their expertise in fitting psychophysics data, and Sanne Kikkert for her expertise in fMRI analysis.

# Additional information

### Competing interests

HJ-B: Reviewing editor, *eLife*. The other authors declare that no competing interests exist.

### Funding

| Funder | Grant reference number | Author |
| --- | --- | --- |
| University College, Oxford | | James Kolasinski |
| Wellcome | 104128/Z/14/Z | Tamar R Makin |
| Wellcome | 102584/Z/13/Z | Charlotte J Stagg |
| Medical Research Council | MR/L009013/1 | Saad Jbabdi |
| Wellcome | 110027/Z/15/Z | Heidi Johansen-Berg |

The funders had no role in study design, data collection and interpretation, or the decision to submit the work for publication.

### Author ORCIDs

James Kolasinski, http://orcid.org/0000-0002-1599-6440
Tamar R Makin, http://orcid.org/0000-0002-5816-8979
John P Logan, http://orcid.org/0000-0002-4469-2948

### Ethics

Human subjects: All data were acquired in accordance with local central university research ethics committee approval (University of Oxford MSD-IDREC-C2-2013-05). Eighteen participants were recruited, each providing written informed consent to take part in this study, and for the results of this study to be published.

## Author contributions

JK, Conception and design, Acquisition of data, Analysis and interpretation of data, Drafting or revising the article; TRM, Conception and design, Analysis and interpretation of data, Drafting or revising the article; JPL, Acquisition of data, Analysis and interpretation of data, Drafting or revising the article; SJ, Analysis and interpretation of data, Drafting or revising the article; SC, Acquisition of data, Drafting or revising the article; CJS, HJ-B, Conception and design, Analysis and interpretation of data, Drafting or revising the article

## Additional files

### Supplementary files

• Supplementary file 1. Consistent patterns of peak-to-peak distance (mm) in cortical z-statistic digit representations. The observed consistency was quantified with Cronbach's α for each digit pair across the three time points. The resulting values support a high degree of consistency in the relative peak-to-peak distance across the observed digit maps over time.

• Supplementary file 2. Summary of goodness of fit between phase-encoding model and fMRI data for each digit and timepoint. These values support a high degree of consistency in the model fit within each participant and digit across the fMRI time points: 181 TRs of fMRI data.

• Supplementary file 3. Demographic information for participants recruited to fMRI and behavioural cohorts. F: female, M: Male, R: right handed.

• Supplementary file 4. Full statistical outputs for repeated measures ANOVAs and simple main effects analyses.

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
