## [Decision Letter]

Thank you for submitting your article "Rapid remapping of human somatosensory cortex" for consideration by *eLife*. Your article has been reviewed by three peer reviewers, and the evaluation has been overseen by Rich Ivry, the Reviewing Editor, and Sabine Kastner as the Senior Editor. The following individual involved in review of your submission has agreed to reveal their identity: Hubert Dinse (Reviewer #1).

The reviewers have discussed the reviews with one another and the Reviewing Editor has drafted this decision to help you prepare a revised submission.

Summary:

Three reviewers have evaluated the paper. We all find the paper intriguing with the potential to make a nice addition to the literature on short-term plasticity in the human somatosensory system. The converging evidence from fMRI and psychophysics has the potential to make a compelling story. However, there are a number of substantive issues that must be addressed in a revision to make the case more convincing.

Essential Revisions:

1) The main issues concern the analyses.

1A) We take it that your key findings, namely that the main reorganization effects are observed in the untreated fingers, were unexpected – that you had anticipated observing changes in the representation of the treated fingers (perhaps in addition to the untreated fingers). Granted, one has to follow the data. However, the situation is puzzling. You report no change in the peak locations, representational areas of all fingers, and overlap of D2-D3 (treated fingers). You do find changes in overlap of D3-D4 (smaller overlap) and D4-D5 (greater overlap). If we put this all together, the results indicate that there is a change in the flank of D4 such that the region near D3 shifts to the other side (near D5). That is, with no change in peak of D4 or its overall area, there should be some sort of marked asymmetry in representation of D4, with much larger flank on side towards D5 (with corresponding reduction in flank on D3 side to keep area constant). Is this correct? Or does the significant overlap shift emerge as a consequence of non-significant changes in finger peaks, areas, etc.? We would like to see a graphical depiction to better understand this puzzling pattern, perhaps some pre-post variant on the graph you show in Figure 1. Ideally, this would be evident in maps of individuals (as you did in your previous J Neurosci paper) as well as in some sort of group analysis. Be sure to clarify how you measured the peaks.

1B) We would like a sense of the variability of individual somatotopic maps. Two measures to consider here. First, a measure of goodness-of-fit between the phase-encoding model and brain activity (either the average R2 or r-value for each digit). Second, a quantitative evaluation of inter-subject stability, perhaps performed by calculating Cronback's α (Cronbach, 1951; Grinband et al., Neuroimage 2008) to measure the correlation between the spatial patterns of digit representations of each subject.

1C) Other authors have shown that the representational structure of the fingers may depict the invariant organization of the somatosensory cortex better than the exact spatial distribution of the finger clusters (Ejaz et al., 2015). Such measures are not prone to some of the problems of the measurements of cortical overlap used here (e.g., dependence on map thresholds. Representational spaces (Rss) can be computed and compared between sessions, allowing an assessment of the distance between the brain activity patterns related to each digit before and after gluing. This would allow you to confirm that the representational structure of the digits is modified in the post-gluing session only, and not in the control conditions.

1D) Please clarify how binary images were obtained to calculate the Dice coefficient. Did the binary maps obtained through different thresholds show the same patterns of overlap across conditions (i.e., those reported in Figure 1)?

1E) Methods suggest that the individual maps of correlation coefficients were averaged and the z-score was calculated on the group level map. Would it be better to transform the single-subject maps to z-scores before averaging them?

1F) Not clear if you have averaged across positive and negative BOLD signals. Would analyzing negative and positive BOLD separately reveal a different picture of the changes?

2) While the psychophysics is impressive, the temporal order task seems to provide an indirect way to assess change in overlap representation. A spatial localization task would provide a more direct test (less confusing between D3 and D4, more confusions between D4 and D5), as well as add to null results of no change in confusions for D2 and D3. This experiment would further bolster your conclusions, especially important here given that the imaging results are unexpected.

3) The methods section is difficult to sort through. A few of us came away from text assuming you had done before/after contrasts, with one contrast Baseline vs. Control and the other Baseline vs. Gluing. Your design figure suggests that you may have planned things this way for decided there were concerns with the first session (familiarity with scanner environment?) and thus developed a different set of pairings. As such, one contrast is separated by 24 hours and the other by 4 weeks. We recognize that, by counterbalancing things, this is not a confound. Indeed, one could say it is even more impressive that there is little change between Control 1 and 2 given that the time between scans is much greater for this contrast (for half the participants) compared to between (one of the) control scans and Gluing scan. However, we do think it important to report if there is an effect of delay. That is, do you get more stability when the control scans are separated by 24 hours vs. 4 weeks (and similarly, more difference with Gluing when there is 4 weeks vs. 24 hours). The n will be quite small (4 vs. 5) in an analysis of order but we think the order issue should be addressed. Given that the small n may lack power in a statistical test, we would also like to see a supplemental figure with the fMRI data for the two orders.

4) While appreciating the novelty of your design, we do believe that there is a substantial body of literature, both human and non-human, on short-term plasticity. You cite the Pleger et al. '03 paper which is most similar to the current work. There are other uncited papers that seem relevant to the discussion, some using fMRI and some using other methods. Please consider relevance of Pleger et al. 2001 (PNAS), Dinse et al. 2003 (Science), Hodzik et al. 2004 (J Neurosci), Tegenthoff et al. 2005 (PLoS Biol), Lissek et al. 2009, Kalisch et al. 2009, and Mogilner et al. 1993 (PNAS).

5) Your method is similar to that used by Merzenich and colleagues in the late 80's. They, of course, looked at long-term changes and observed dramatically different results. A (slightly) expanded discussion of these differences and possible mechanisms would better situate your paper in the somatosensory reorganization literature.

---

## [Author Response]

[…]

*Essential Revisions:*

*1) The main issues concern the analyses.*

*1A) We take it that your key findings, namely that the main reorganization effects are observed in the untreated fingers, were unexpected – that you had anticipated observing changes in the representation of the treated fingers (perhaps in addition to the untreated fingers). Granted, one has to follow the data. However, the situation is puzzling. You report no change in the peak locations, representational areas of all fingers, and overlap of D2-D3 (treated fingers). You do find changes in overlap of D3-D4 (smaller overlap) and D4-D5 (greater overlap). If we put this all together, the results indicate that there is a change in the flank of D4 such that the region near D3 shifts to the other side (near D5). That is, with no change in peak of D4 or its overall area, there should be some sort of marked asymmetry in representation of D4, with much larger flank on side towards D5 (with corresponding reduction in flank on D3 side to keep area constant). Is this correct? Or does the significant overlap shift emerge as a consequence of non-significant changes in finger peaks, areas, etc.? We would like to see a graphical depiction to better understand this puzzling pattern, perhaps some pre-post variant on the graph you show in Figure 1. Ideally, this would be evident in maps of individuals (as you did in your previous J Neurosci paper) as well as in some sort of group analysis. Be sure to clarify how you measured the peaks.*

As the reviewers suggest, the pattern of changes was unexpected in light of previous work in non-human primates. However, we feel that while the results were not perhaps those we might have expected, it should be emphasised that human hand use differs considerably in terms of the dexterous abilities. This could in turn drive compensatory strategies in humans, which were not seen in monkeys: for example, adapting usage patterns of different non-glued digits to achieve the daily tasks. As such, the triggers for reorganization in this study may be different from those in primate studies.

While unexpected, our findings have been bolstered by both additional analysis of independent fMRI data (point 1C), as well as complementary evidence from psychophysics, including additional data and analysis (point 2).

Following the reviewers’ suggestion, we have now provided additional figures exploring the shift in digit 4 at the level of individual participants (Figure 2). In brief, we show that compared to both control conditions, the representation of digit 4 in the glued condition shows an expansion at the flank adjacent to digit 5, and a partial contraction at the flank adjacent to digit 3. This pattern is shown for all nine participants in the fMRI experiment, including comparison of the glued condition to both of the control conditions in all cases. The location of peak vertices are also included, demonstrating no consistent pattern of movement along the axis of different digits, compatible with the stable peak-to-peak distances presented in [Supplementary-material SD5-data]. We believe that this pattern of expansion is in keeping with the short-term nature of the observed plasticity, which is likely underpinned by changes in the activity of pre-existing connections in the boundary regions between digit representations.

We have included additional references to the figure in the main text, and refined our discussion of the results to explain in qualitative terms the nature of an expansion and contraction at the flanks of the representation of digit 4, and the likely mechanistic underpinnings of such a change. We have also included details of how the peaks were defined, and the distances between these peaks were calculated.

Results:

“In order to more fully explore the observed change in inter-digit overlap with no change in overall cortical surface area of each digit representation, the fMRI representations of digit four used to calculate the degree of cortical overlap between adjacent digits (Dice coefficients) were visualized for each participant, at each session (Figure 2). This data revealed at the level of individual participants’ digit maps that the observed changes presented in Figure 1 were driven by an expansion in the representation of digit 4 adjacent to digit 5, and a corresponding contraction at the boundary with digit 3.”

Discussion:

“The observation of a cortical shift in the digit 4 (Figure 2), rather than an overall enlargement of the representation, distinguishes the observed pattern of cortical change from previous studies of artificial tactile co-activation, wherein a generalized cortical magnification of the manipulated digit is observed (Pleger et al., 2001; Pleger et al., 2003; Hodzic et al., 2001). The cortical shift perhaps reflects how naturalistic behavioural changes in hand use are reflected in the cortex, with a remapping based on functional need; in this case leading to perceptual improvements in certain digit pairs (D3/D4), and worsening in others (D4/D5).”

Methods: Phase-encoding fMRI analysis: inter-digit overlap

“Patterns of peak activation for each digit during each session were calculated using Connectome Workbench, defined as the vertex with the peak *z*-statistic for each digit within the FreeSurfer-derived SI region of interest. The geodesic distance between adjacent digit peaks was subsequently calculated ([Supplementary-material SD5-data]).”

*1B) We would like a sense of the variability of individual somatotopic maps. Two measures to consider here. First, a measure of goodness-of-fit between the phase-encoding model and brain activity (either the average R2 or r-value for each digit).*

We are happy to provide a measure of the goodness of fit between the phase-encoding models and the BOLD fMRI signal within the FDR thresholded digit representations of each participant acquired in each session. These cross-correlations represent the goodness of fit between the model and BOLD signal across 181 TRs (TR = 1500 ms). This data is provided in [Supplementary-material SD6-data].

In brief, we demonstrate that there is no systematic change in the goodness of fit across the different sessions, therefore discounting the possibility of this account for the observed changes in the glued condition.

Results:

“No systematic difference in the fit between the phase-encoding models and fMRI signal were observed across sessions ([Supplementary-material SD6-data]).”

*Second, a quantitative evaluation of inter-subject stability, perhaps performed by calculating Cronback's α (Cronbach, 1951; Grinband et al., Neuroimage 2008) to measure the correlation between the spatial patterns of digit representations of each subject.*

We understand from this comment that the reviewers would like to see some quantification of the degree of inter-individual variability in spatial distribution of digit maps across individual participants. As we recently demonstrated using the same phase-encoding mapping technique and scan parameters (Kolasinski et al. J Neurosci 2016; Figure 5B), the degree of intra-individual variability in the digit maps across time is low, however there is a considerable degree of variability in the spatial distribution of digit representations across different participants. While some individuals show a high concordance in the representation of the same digits, the majority do not. This is supported by evidence of striking variability in the morphology and angle of activation patterns across different participants (Kolasinski et al. J Neurosci 2016; Figure 3 and Figure 4). As such, we felt that a group-level analysis co-registering the fMRI data would not be suitable to identify the more subtle effects of the gluing manipulation.

However, if the reviewers are referring to some additional aspect of inter-individual variability, we would of course be happy to provide any additional data or analysis that would be of use in evaluating this manuscript.

*1C) Other authors have shown that the representational structure of the fingers may depict the invariant organization of the somatosensory cortex better than the exact spatial distribution of the finger clusters (Ejaz et al., 2015). Such measures are not prone to some of the problems of the measurements of cortical overlap used here (e.g., dependence on map thresholds. Representational spaces (Rss) can be computed and compared between sessions, allowing an assessment of the distance between the brain activity patterns related to each digit before and after gluing. This would allow you to confirm that the representational structure of the digits is modified in the post-gluing session only, and not in the control conditions.*

We agree with the reviewers that multivoxel pattern analysis approaches, including analysis using representational spaces, do offer many benefits in terms of assessing the representational structural of a given cortical region. We feel that this is specifically true here, given the strong hypotheses we have regarding remapping from the phase-encoding fMRI and TOJ psychophysics results. Ejaz et al. 2015 have shown this approach to be effective in the assessment of digit representations in S1. We have therefore included an additional analysis using a similar representational similarity analysis (RSA) approach to assess the representational distances between different digit representations across the three experimental sessions under study.

This analysis uses block-design task fMRI data acquired during each session of the experiment. It therefore provides a valuable replication of the observed remapping, independent of the previously discussed analysis of phase-encoding task fMRI data.

Below we have provided the additional descriptions of RSA methods, analysis, and results in the manuscript. In brief, we first find that RSA yields representational dissimilarity matrices comparable to those previously presented by Ejaz et al. 2015 in S1, showing greater representational similarity of adjacent digits, specifically of digits which show common synergistic usage patterns. Analysis of the distance metrics between adjacent digit pairs (D2/D3, D3/D4 and D4/D5) across the three experimental sessions reveals a significant increase in the representational distance between D3 and D4, and a significant reduction in the representational distance between D4 and D5 (Figure 3). No change is observed between D2 and D3, the manipulated digits. These results have been subject to multidimensional scaling for intuitive graphical representation (Figure 3).

Methods:

“Block design task

A block task was undertaken involving the same cued movements of a given digit in blocks of 12 seconds, contrasted with 12-second blocks of rest. Blocks of movement for each digit were repeated four times, giving a total of 16 movement blocks and 16 rest blocks. The total duration of the task was 6 minutes 30 seconds. Block order was counterbalanced, and different orders used across the different visits, counterbalanced across participants.*”*

“Block task fMRI: representational similarity analysis

Multivariate analysis was conducted using block task data in order to compare the representational structure of digit maps in the glued versus control sessions. Block task fMRI data were analysed at the single-subject level using a standard GLM approach using FSL FEAT (Jenkinson et al., 2011), with an individual regressor for movement of each digit (D2, D3, D4, D5), with a γ-HRF boxcar (four 12-second blocks of activity per digit during the task). […] For visualization purposes, the individual participant representational distance matrices (RDMs) for each session was subjected to classical multidimensional scaling (MDS). Within each session, the result of the MDS for each participant was subject to procrustes alignment and correct of standard errors, as described previously (Ejaz et al., 2015).”

Results:

“Representational similarity analysis conducted on independent block-design fMRI data acquired during each session also implicated changes in the representation of digit 4 (Figure 3). These data demonstrated a shift in the representation of D4 away from D3, and towards D5.”

*1D) Please clarify how binary images were obtained to calculate the Dice coefficient.*

We apologise that this process was not sufficiently clear in the Methods section, and have clarified this in the main text. In summary, for each participant and each session, a *z*-statistic image was projected from 3D volumetric space onto the 2D cortical surface reconstruction of the specific participant. Once projected to the cortical surface, the 2D maps of each digit for each participant and each session were subject to thresholding using FDR (α=0.01). The resulting thresholded *z-*statistic 2D maps were then binarised, and used in the calculation of Dice coefficients. We have clarified this as follows in the Methods section:

Methods: Phase-encoding fMRI analysis: inter-digit overlap

“Cortical surface reconstructions were derived from MEMPRAGE images using FreeSurfer (Marcus et al., 2011). […] Where A and B are the area of the two digit representations, the Dice Coefficient is expressed as:”

*Did the binary maps obtained through different thresholds show the same patterns of overlap across conditions (i.e., those reported in Figure 1)?*

We selected FDR α=0.01 as a threshold as this was previously applied in our analysis of overlap between adjacent digits ((Kolasinski et al., 2016); Figure 6). We feel that given the same scan parameters and analysis were applied here, a consistent approach to thresholding is justified. More stringent thresholds (e.g. FDR α=0.001) do not yield any overlap between adjacent representations. We have re-run the analysis with a more lenient threshold (FDR α=0.05), which shows the same general pattern of change, but with an expected increase in variance.

*1E) Methods suggest that the individual maps of correlation coefficients were averaged and the z-score was calculated on the group level map. Would it be better to transform the single-subject maps to z-scores before averaging them?*

We have attempted to clarify this point in the Methods section. Consistent with previous applications of phase-encoding cross-correlation analysis (Orlov et al., 2010) for each participant’s data in each session, we averaged the r-value maps from the forward and backward variants of the phase-encoding task, to avoid order effects on the resulting activation (e.g. D3 always following D2 in the forward variant). The resulting *r-*value maps for each participant and session were then used to construct *z-*statistic maps of individual digit representations for that participant and session. At no point were the *r*-value maps averaged across participants or across sessions (i.e. at the group level). We have clarified this approach in the text as follows:

Methods: Phase-encoding fMRI analysis: inter-digit overlap

“Each iteration of the model was assigned to a specific digit based on the position of the ‘on’ period in the digit cycle. […] The forward and backward*r*-value maps were then averaged to yield a digit representation for each participant in each session. A corresponding *z*-statistic map was calculated for each of the four digit representations in each participant, in each session.”

Methods: MRI Data Preprocessing

“fMRI data were processed using tools from the FMRIB Software Library (Jenkinson et al., 2011). All fMRI data were analysed at the level of individual participants and sessions; no group averaging was undertaken across session or across participants.”

*1F) Not clear if you have averaged across positive and negative BOLD signals. Would analyzing negative and positive BOLD separately reveal a different picture of the changes?*

The analysis herein has not considered the positive and negative components of the BOLD signals independently. Modelling of negative BOLD considers signal changes relative to baseline. The phase-encoding fMRI task does not have periods of baseline, and it modelled with a travelling wave, rather than a GLM. As such, this data is not suited to extracting specific information regarding changes in the negative BOLD signal. Similarly, the representational similarity analysis (point 1C above), also does not independent consider changes in negative and positive BOLD signals.

While considering changes in negative BOLD independently could provide complementary information in this manuscript, we feel that this analysis is outside the scope of the main hypothesis, and would not provide an easily interpretable additional measure in this case. We are concerned that by including this analysis, particularly in a short report manuscript, the focus of the work may become unclear to potential readers, particularly given the focus of negative BOLD in studies of motor rather than somatosensory cortex. We therefore feel that this would be best addressed separately in our future work. However, given the broadening interest in negative BOLD in sensorimotor cortex (Zeharia et al., 2012), we do feel that this should be mentioned as an area of potential interest. We have included a reference to this in the Discussion as follows:

Discussion:

“The recent observation of somatotopic information in the negative BOLD signal in primary motor cortex also raises questions regarding the mechanisms maintaining cortical topographic organisation, and whether such signals show the same propensity for plastic change as seen in this study (Noa Zeharia, 2012), though the negative BOLD signal was not considered independently herein.”

*2) While the psychophysics is impressive, the temporal order task seems to provide an indirect way to assess change in overlap representation. A spatial localization task would provide a more direct test (less confusing between D3 and D4, more confusions between D4 and D5), as well as add to null results of no change in confusions for D2 and D3. This experiment would further bolster your conclusions, especially important here given that the imaging results are unexpected.*

We agree that it is important to support our imaging results with sound behavioural results. We selected the temporal order judgment task because it engages the representations of two adjacent digit representations in the cortex, and therefore provides a window on the extent of their shared territory, which is highly comparable to the measures of interest from our imaging results.

A spatial localisation task, in which participants are asked to judge the location of a tactile stimulus, could provide a complementary behavioural measure. We would hypothesise that the gluing manipulation would induce more mislocalisations between the digits where cortical overlap has increased.

However, we have concerns regarding the feasibility of a spatial localisation task for this study. The duration of testing required to record a relatively small number of errors made such a task undesirable (Schweizer et al., 2001; Braun et al., 2005). We feel that this task is perhaps better suited in patient populations, where the number of mislocalisations is expected to be increased due to somatosensory dysfunction. We instead used an active task, which invited a greater number of errors.

In this work we posit that usage-dependent plasticity is responsible for the observed shift in the representation of digit 4, as it is freed from anatomical enslavement to digit 3, and free to function even more synergistically with digit 5. We have now included additional data acquired as part of the behavioural experiment, which speaks to the question of mislocalisation from a motor perspective. Specifically, after the gluing manipulation, is there evidence for an increased synergy in digit 4 and 5, in the form of increased confusion in movements between them? We acquired data during a rapidly-cued button-press task, which could address to the concept of mislocalisation.

We have included details of the task, analysis, and results below. In brief, we find that in keeping with our hypothesis, there is an increased level of confusion of movements between digit 4 and digit 5 after the gluing manipulation, compared with the two control conditions. No such reduction is observed between digit 3 and digit 4. Nonetheless, we feel that this provides corroborative behavioural evidence supporting the observed pattern of remapping. Moreover, in contrast to a sensory localisation task, we feel that this provides additional insight to data from the TOJ task, demonstrating potential differences in movement synergies after the glued manipulation that may explain the somatosensory changes observed.

Methods:

“Motor confusion task

A motor confusion task was undertaken to behaviourally assess whether the gluing manipulation induced changes in motor function, which aligned with the observed patterns of somatosensory remapping in experiment 1, and any tactile perceptual changes from the temporal order judgment task. […] Analysis was conducted on the number of mis-presses made across adjacent digit pairs in either direction (D2/D3 & D3/D2, D3/D4 & D4/D3, D4/D5 & D5/D4).”

Results:

“In other words, after the gluing manipulation, tactile discrimination was improved across D3-D4, and was diminished across D4-D5, with no change in the glued digits D2-D3. […] Data from this task demonstrated an increase in the level of confusion between digits 4 and 5 after the gluing manipulation compared with the control conditions (Figure 4—figure supplement 1).”

*3) The methods section is difficult to sort through. A few of us came away from text assuming you had done before/after contrasts, with one contrast Baseline vs. Control and the other Baseline vs. Gluing. Your design figure suggests that you may have planned things this way for decided there were concerns with the first session (familiarity with scanner environment?) and thus developed a different set of pairings.*

We have attempted to clarify the Methods section to explain these important elements of the experimental design. We hope that by adding this detail, the Methods section will with be easier to follow. The study design figure has been modified to reflect the fact that control 0 was not included in any analysis, and rather acted as a habituation scan as the reviewers suggest, due to the frequent presence of excessive motion during this scan session, attributed to the novel scanner environment at 7 tesla.

Methods: fMRI experiment

“Nine healthy adult volunteers participated in this experiment, in line with the closest previous studies of longitudinal remapping in human SI (Pleger et al., 2003; Young, 2003). […] The order in which the participants undertook the control and gluing condition was counterbalanced across the group.”

*As such, one contrast is separated by 24 hours and the other by 4 weeks. We recognize that, by counterbalancing things, this is not a confound. Indeed, one could say it is even more impressive that there is little change between Control 1 and 2 given that the time between scans is much greater for this contrast (for half the participants) compared to between (one of the) control scans and Gluing scan. However, we do think it important to report if there is an effect of delay. That is, do you get more stability when the control scans are separated by 24 hours vs. 4 weeks (and similarly, more difference with Gluing when there is 4 weeks vs. 24 hours). The n will be quite small (4 vs. 5) in an analysis of order but we think the order issue should be addressed. Given that the small n may lack power in a statistical test, we would also like to see a supplemental figure with the fMRI data for the two orders.*

As requested, we have included a summary of the data presented in Figure 1, split according to the different order of sessions, in Figure 1—figure supplement 3. As can be seen, the pattern of the results was similar across the two sub-cohorts, suggesting that the overall effect is not being driven by the differing relative timing of the glued condition relative to the two control conditions.

*4) While appreciating the novelty of your design, we do believe that there is a substantial body of literature, both human and non-human, on short-term plasticity. You cite the Pleger et al. '03 paper which is most similar to the current work. There are other uncited papers that seem relevant to the discussion, some using fMRI and some using other methods. Please consider relevance of Pleger et al. 2001 (PNAS), Dinse et al. 2003 (Science), Hodzik et al. 2004 (J Neurosci), Tegenthoff et al. 2005 (PLoS Biol), Lissek et al. 2009, Kalisch et al. 2009, and Mogilner et al. 1993 (PNAS).*

Thank you for providing these additional citations. As this manuscript was initially prepared as a short report, we were conscious of limiting the depth of our discussion. However, given the suggested publications would improve the context of the reported findings, we are of course happy to provide a further discussion of this work, and how it relates to our findings herein. We have integrated these citations across the Introduction and Discussion, as follows:

Introduction:

“In the human brain there has also been evidence of experience dependent remapping in SI. […] However, only limited longitudinal evidence exists that directly investigates remapping at the level of human SI, either in response to altered hand usage patterns (Stavrinou et al., 2007), or more intensive Hebbian co-activation paradigms delivering specific patterns of tactile stimulation to the fingertips (Pleger et al., 2001; Pleger et al., 2003; Hodzic et al., 2001; Vidyasagar, Folger and Parkes, 2014).”

Discussion:

“The observation of a cortical shift in the digit 4 (Figure 2), rather than an overall enlargement of the representation, distinguishes the observed pattern of cortical change from previous studies of artificial tactile co-activation, wherein a generalized cortical magnification of the manipulated digit is observed Pleger et al., 2001; Pleger et al., 2003; Hodzic et al., 2001). The cortical shift perhaps reflects how naturalistic behavioural changes in hand use are reflected in the cortex, with a remapping based on functional need; in this case leading to perceptual improvements in certain digit pairs (D3/D4), and worsening in others (D4/D5).”

*5) Your method is similar to that used by Merzenich and colleagues in the late 80's. They, of course, looked at long-term changes and observed dramatically different results. A (slightly) expanded discussion of these differences and possible mechanisms would better situate your paper in the somatosensory reorganization literature.*

We agree that given the contrast between our short time findings, and the longer-term remapping studies of Merzenich and colleagues, a short additional discussion of the possible mechanisms would be warranted. We have modified the original Discussion to better integrate content on Merzenich’s work, as well as more recent work that may shed light on the likely underlying mechanisms of plasticity.

Discussion:

“The 24-hour manipulation in this study may have promoted more adaptive behavioural strategies using unaffected digits, particularly in humans, whose hand usage and dexterous repertoire both exceed those of new world monkeys (Young, 2003). […]While BOLD fMRI is unable to map at a comparable spatial resolution to such studies, it is conceivable that the observed changes in cortical activation could be underpinned by shifts in population receptive fields at the boundary of cortical digit representations (Besle et al., 2013), or even changes in multidigit receptive fields (Thakur, Fitzgerald and Hsiao, 2012).”

“The question remains: what drives the observed changes in the cortex? Given the potential role for adaptive or compensatory behaviour as a driving force for the observed cortical remapping, bottom-up signals are likely to play a significant role in any receptive field changes that underpin remapping. […] The recent observation of somatotopic information in the negative BOLD signal in primary motor cortex also raises questions regarding the mechanisms maintaining cortical topographic organisation, and whether such signals show the same propensity for plastic change as seen in this study (Noa Zeharia, 2012), though the negative BOLD signal was not considered independently in this study.”